# A Real-Time Measurement-Modeling System for Ship Air Pollution Emission Factors

Fan Zhou [1,2,*] , Jing Liu [1,2], Hang Zhu [3], Xiaodong Yang [3] and Yunli Fan [1,2]

1   College of Information Engineering, Shanghai Maritime University, Shanghai 201306, China; jingliu_cv@163.com (J.L.); yunlifan_cv@163.com (Y.F.)
2   Shanghai Engineering Research Center of Ship Exhaust Intelligent Monitoring, Shanghai 201306, China
3   Sanxia Maritime Safety Administration of People's Republic of China, Yichang 443002, China; hangzhu_sx@yeah.net (H.Z.); xiaodongyang_sx@yeah.net (X.Y.)
*   Correspondence: fanzhou_cv@163.com

**Abstract:** The lack of techniques for monitoring ship emissions all day and in all weather conditions to obtain real-time emission factor values is the main problem in understanding the characteristics of ship emissions, and there is still no perfect solution. In this study, a real-time measurement-modeling system was designed and implemented. The system was divided into three parts: (1) a portable exhaust monitoring device, which could be mounted on a drone, aircraft, patrol boat, dock, and bridge crane, as well as on the shore, to conduct all-weather and real-time online monitoring of ship emissions; (2) a monitoring information platform for ship emissions, based on a Spring + Spring MVC + MyBatis (SSM) framework and Vue front-end technology; and (3) a cloud server that received real-time ship emission measurement data and stored it after verification and analysis to calculate the pollutant gas and particulate matter emission factors. Following development, this system was used to monitor the emissions of ocean-going and inland river ships. Analysis of the acquired data showed that the system could effectively measure the emission factors of ship exhausts full-time in a variety of weather scenarios. This system can improve the efficiency of maritime law enforcement and provide technical support for promoting the construction of ship emission control areas. It can also help researchers obtain ship emission data, as well as an improved understanding of the emission characteristics of ships.

**Keywords:** air pollution; ship emissions; emission factor; environmental monitoring; real-time monitoring; SSM framework

## 1. Introduction

As the global economy has developed in recent decades, the shipping industry has played an increasingly important role in international trade [1]. As a result, more attention is being paid to air pollution caused by ship emissions [2–5]. The sulfur and nitrogen oxides as well as particulate matter (PM) emitted by ship exhausts harm human health and the environment [6,7]. At present, international conventions and regulations of relevant regions and countries have increased the stringency of emission requirements regarding atmospheric pollutants, especially sulfur oxides, from ships. Regulations on marine fuel sulfur content have been continuously issued, revised, and implemented [8–12].

Many researchers have examined ship emissions, taking actions such as compiling emission inventories [5,13], improving knowledge of the harm of ship emissions [6,14], and developing ship emission reduction strategies [15,16]. Among these, it is particularly important to measure the emission factors involved in ship emissions [7]. Emission factors can be used to compile a ship emission inventory and then evaluate the impact of those emissions, and they can also be used by maritime regulatory authorities to check the basis of excessive discharge or sulfur content in fuel [17,18]. Therefore, developing methods to measure the emission factors of ship exhaust gas and PM is an important research goal.

In 2005, MARPOL 73/78 Annex VI, enacted by the International Maritime Organization (IMO), officially came into force. This opened the gates for researchers to study and estimate the emission factors of exhaust gases, PM, and fuel sulfur content (FSC) through monitoring ship emissions. During 2006 and 2007, the Identification of Gross-Polluting Ships (IGPS) measurement system was built [19–21]. The remote surveillance of ship emissions has demonstrated that enforcing IMO emission legislation can be conducted using the sniffer and optical techniques. During 2007–2013, the Shipping-Induced $NO_x$ and $SO_x$ Emissions—Operational Monitoring Network (SNOOP) project was designed to measure and characterize the emissions of regularly operating ships equipped with different after-treatment systems under winter and summer conditions in two different environments in Finland [22,23]. In September 2009, the techniques available for measuring ship emissions were reviewed by the European Commission's Joint Research Centre and tested during a measurement campaign in the harbor of Rotterdam [24]. Between 2014 and 2016, compliance monitoring for the MARPOL Annex VI (CompMon) Coalition was established between the European Union (EU) member states [25,26]. The name of the coalition's project report was the "MARPOL Annex VI Best Practice Monitoring" report, which aimed to enhance efficiency and safety during monitoring operations by providing information on flight approach, altitude, distance to ships, speed, plume localization, sampling attempts, weather minima, and safety recommendations.

These results provide a key basis for obtaining ship emission characteristics. However, these studies were based on the purpose of their research, rather than practical applicability. To effectively promote the monitoring of ship emission control areas (ECAs), a stable, all-weather, and all-day monitoring system is required, and relevant researchers and maritime regulatory authorities need real-time access to this information. Local-scale online sensor data integrated with measuring models could be a valid tool to overcome the issues of irregularity and limited spatial representativeness of standard fixed monitoring stations by exploiting the spatial coverage allowed by models.

In recent years, the rapid development of new technologies such as Big Data, the Internet of Things (IoT), and cloud computing provides a technical foundation for real-time monitoring of ship emissions or air quality. In the work of Merico et al., researchers developed a system based on the integration of measurements collected using a network of low-cost online sensors with local-scale dispersion modeling. It was implemented, as a pilot action, using the information and management software of the Bari harbor. It operated in near-real-time, in forecast mode, and on archived data for long-term assessments [27]. Christos et al. exploited a unique business window of opportunity for a platform architecture that harmonized, through Big Data technologies, data collected from various sensors onboard. It implemented large-scale processing techniques to support decision-making procedures in shipping [28]. Spandonidis et al. proposed a new, low-cost, compact, and modular IoT platform for air quality monitoring in urban areas that comprised dedicated, low-cost, low-power hardware and the associated embedded software to enable the measurement of particulate ($PM_{2.5}$ and $PM_{10}$), NO, CO, $CO_2$, and $O_3$ concentration in the air, along with relative temperature and humidity [29].

These studies provide the technical basis for better monitoring of ship emissions but tend to focus on software or hardware rather than well-developed solutions. Meanwhile, maritime supervision has its own particularities, which are different from those of general scientific research, and has higher requirements for real-time data collection and data stability [30]. A comprehensive solution is needed to obtain ship emissions all day and in all weather conditions with real-time emission factor values. For this purpose, we developed a ship exhaust monitoring system, which includes an integrated software solution, hardware, and calculation methods. This system provides a real-time measurement-modeling approach for monitoring the air pollution emission factors of ships.

The rest of the paper is organized as follows. The system overview diagram is introduced in Section 2, in which data acquisition equipment is listed in Section 2.1, software architecture in Section 2.2, data analysis and understanding in Section 2.3, and application

in Section 2.4. The experiments, results, and discussion are given in Section 3. Finally, some conclusions are provided in Section 4.

## 2. Materials and Methods

The current experiments in monitoring ship emission are primarily oriented towards research or isolated monitoring activities rather than comprehensive monitoring, which does not meet the needs of maritime authorities for emission monitoring and supervision. Therefore, we designed portable exhaust monitoring equipment which could be deployed in a variety of areas such as on drones, aircraft, patrol boats, docks, and bridge cranes. Ship emissions were monitored in real-time and in all weather conditions, and measurement data were sent to a cloud computing center. After processing and analysis, the data were used to provide information services and decision support to local maritime law enforcement personnel. The data could be sent to data centers for storage and further analysis, or be used as data sources for scientific research. Following this approach, a monitoring network model was built consisting of grid points + suspicious target tracking + multiple data fusion + spatial–temporal data analysis.

The application scenario of this real-time measurement-modeling system is shown in Figure 1, and is divided into three parts:

1. A portable exhaust monitoring device was designed, which can be mounted on drones, patrol boats, docks, bridge cranes, or other maritime equipment, to carry out all-weather and real-time online monitoring of ship emissions. The device used active suction to measure the content in the exhaust gas, and sent the detection results to a cloud server using 4G transmission.
2. Based on the Spring + Spring MVC + MyBatis (SSM) framework, and using Vue front-end technology, a data monitoring information platform was developed. This software was deployed on a cloud server to receive the monitoring data sent by the device and to provide information services to users.
3. The cloud server received the data, which were verified and parsed, and then used it to calculate the pollutant gas and PM emission factors. The FSC, which is helpful for improving supervision efficiency, was calculated by analyzing the measured data. Maritime law enforcement could therefore preliminarily judge whether the target ship was using fuel with excessive sulfur content without boarding the ship to extract fuel samples.

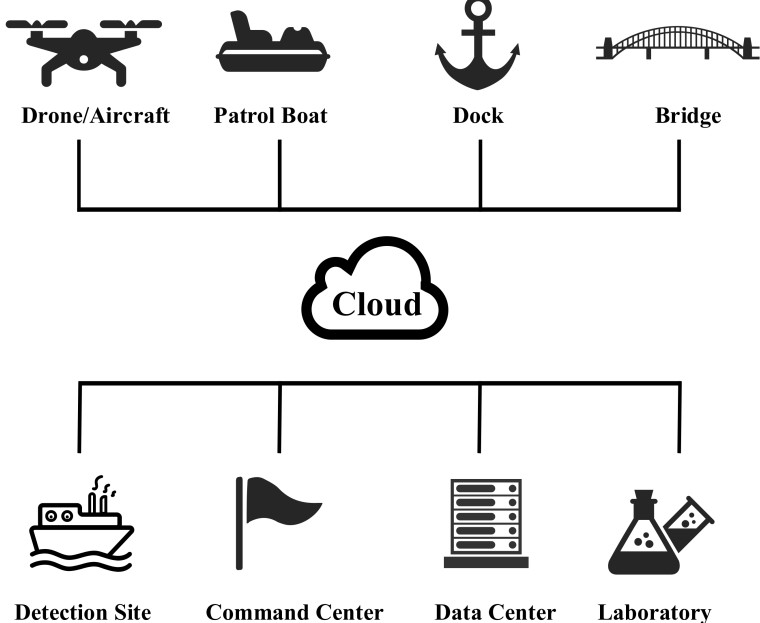

**Figure 1.** System overview diagram.

### 2.1. Data Acquisition Equipment

Monitoring equipment for ship emissions must be capable of all-weather and all-day monitoring performance, but must also account for a variety of installation locations and direct exhaust monitoring, which requires humidity resistance and low power consumption. As shown in Figure 2, the hardware consisted of a solar power system, gas sampling port, dehumidifying filter, micro air pump, gas and particulate matter sensors, sensor PCB package, power supply, STM32 core board, communication module, GPS antenna, and GPRS antenna.

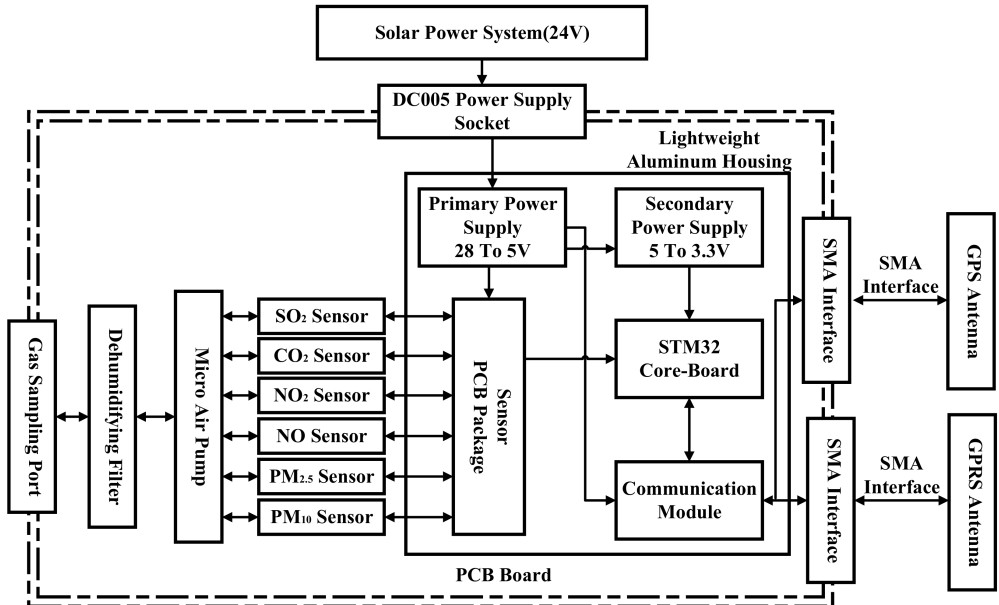

**Figure 2.** Hardware structure diagram.

When the equipment was in operation, the ship exhaust was pumped into the port using a miniature air pump. After dehumidification filtration, the gas ($CO_2$, $SO_2$, NO, $NO_2$) and PM ($PM_{2.5}$ and $PM_{10}$) sensors detected the processed gas through the sampling module and then sent concentration data to the main control processing module via the communication module.

In some ship emission monitoring studies [21,25,26], the sensors for ship emissions used mainly ultra-violet (UV) fluorescence for $SO_2$ and chemiluminescence for $NO_x$ and two other techniques, namely cavity ring-down spectroscopy (CRDS) and non-dispersive infrared (NDIR) absorption for $CO_2$. In this study, sensors were selected with consideration of the convenience and low power consumption; information on the selected sensors is listed in Table 1. The sensor range was determined according to the monitoring environment. In the case of being far away from the ship or with low target emissions, the low range sensor was selected. Otherwise, a high range sensor was chosen. The ranges in Table 1 are the general case choices.

**Table 1.** Sensor parameters of the device, full-scale (FS).

| Sensor | Principle | Range | Accuracy |
|---|---|---|---|
| $SO_2$ | Electrochemistry | 0–2 ppm | ±5% FS |
| $CO_2$ | Non-dispersive infrared | 0–5000 ppm | ±3% FS |
| $NO_2$ | Electrochemistry | 0–2 ppm | ±5% FS |
| NO | Electrochemistry | 0–2 ppm | ±5% FS |
| $PM_{2.5}$ | Laser scattering | 0–1.0 mg/m$^3$ | ±5% FS |
| $PM_{10}$ | Laser scattering | 0–10 mg/m$^3$ | ±5% FS |

These data and GPS data were sent to the cloud server by GPR and GPRS antenna, respectively. The device ran on a solar power system. Therefore, it had low power consumption, a small volume, and a high monitoring accuracy, which are suitable for monitoring ship exhaust gas in the maritime supervision environment.

### 2.2. Software Architecture

The SSM framework was a relatively mainstream Java EE enterprise open-source framework, suitable for building various large-scale enterprise application systems. The SSM framework had the advantages of (1) low cost, (2) savings on development time, (3) good scalability, and (4) good maintainability. Therefore, this study chose the SSM framework to build a server-side program. The software architecture design was based on the SSM framework (Figure 3). The data acquisition layer performed the function of receiving exhaust measurement data from the monitoring equipment. The data processing layer included the validation, parsing, and storage of data, as well as the users' feedback query of the ship's emissions information, which was sent to the data display layer. Statistics, filtering, extraction, and other such data analysis modules were simultaneously set up in this layer to provide information services that could be used for scientific research or maritime supervision. MyBatis used annotations to interact with the database, and performed the operation of adding, deleting, modifying, and checking of data. The front-end program was designed using the Vue framework, which had the advantages of componentization, lightweight code, and fast processing speed. These programs ran in cloud services and could be distributed to different cloud computing nodes according to the specific business volume.

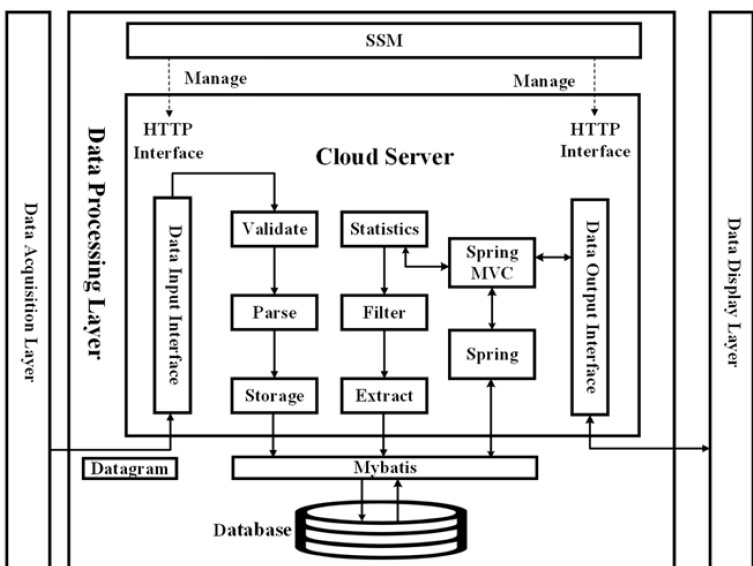

**Figure 3.** Software architecture diagram.

### 2.3. Data Analysis and Understanding

On the cloud server side, the data were analyzed and processed, primarily for calculating the emission factors of gases and PM, as well as the FSC. Currently, there are two main categories of emission factors: fuel-based (g/kg) and power-based (g/kWh). The fuel-based factors were calculated by measuring the concentration of carbon dioxide and other pollutants [21,31]. The power-based factor required real-time access to the ship's energy consumption [7]. Therefore, our equipment primarily measured the fuel-based emission factors, and when the fuel consumption rate was acquired. The fuel-based and power-based emission factors were interconverted [32]. The fuel-based emission factors were calculated using the carbon balance method (Equation (1)). $ER_X$ (emission ratio) is

defined as a ratio of the of the excess concentration of $X$ emitted from a source divided by the excess concentration of $CO_2$ emitted by the source:

$$ER_X \ = \ \frac{X_{peak} \ - \ X_{bg}}{CO_{2,peak} \ - \ CO_{2,bg}} \ = \ \frac{\Delta X}{\Delta CO_2} \tag{1}$$

where $X_{peak}$ is the peak measured value of gas ($SO_2$, NO, $NO_2$) or mass concentration ($PM_{2.5}$, $PM_{10}$), $CO_{2,peak}$ is the $CO_2$ concentration, and $X_{bg}$ and $CO_{2,bg}$ are background concentrations.

The response time for the various sensors was not consistent. Therefore, we used the integral of the measured values for the calculations. The gases were measured in ppm, and PM concentration in μg m$^{-3}$. The emission factor, $EF_X$ (g kg$^{-1}$), is the amount of compound X released per amount of fuel burned, expressed as:

$$EF_X \ = \ \begin{cases} ER_X \ \times \ \frac{M_X}{M_{CO_2}} \ \times \ EF_{CO_2}, \text{ for } SO_2, \text{ NO, } NO_2 \\ ER_{PM} \ \times \ EF_{CO_2}, \text{ for } PM_{2.5}, \ PM_{10} \end{cases} \tag{2}$$

where $M_X$ and $M_{CO_2}$ are the molar masses of gas X and $CO_2$, and $EF_{CO_2}$ is the emission factor of a combusted reference $CO_2$ species (3107 g kg$^{-1}$) [33]. The units of $ER_X$ are ppm/ppm for gases, and ug m$^{-3}$/ug m$^{-3}$ for PM if the $CO_2$ concentration is converted to a mass unit by the ideal gas law at normal temperature and pressure conditions (T = 293.15 K). The FSC was calculated as follows:

$$FSC \ = \ \frac{S(\text{kg})}{fuel(\text{kg})} \ = \ \frac{(SO_{2,peak} \ - \ SO_{2,bg}) \ \times \ A(S)}{(CO_{2,peak} \ - \ CO_{2,bg}) \ \times \ A(C)} \ \times \ 87(\%) \tag{3}$$

where FSC is converted by the emission factor of $SO_2$.

### 2.4. Application

According to the technical principle discussed, we developed the hardware equipment as shown in Figure 4a. It could be mounted on a drone, aircraft, patrol boat, dock, bridge crane, and other marine equipment, to carry out all-weather and real-time online monitoring. For reference, Figure 4b shows the specific location of equipment on a bridge in the Waigaoqiao Port area. The installation position of the equipment collected the exhaust gas of ships passing under the bridge, and ran continuously. The FSC of ships in the monitoring area was obtained and calculated without personnel on-site.

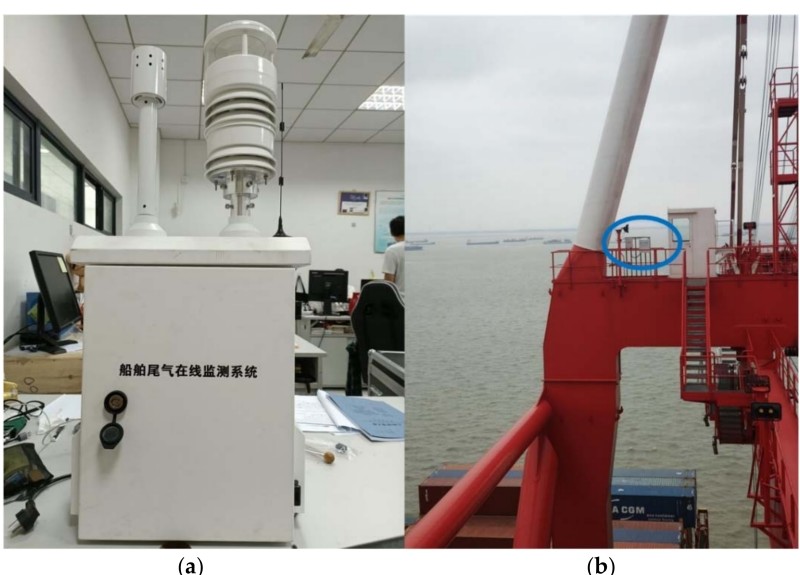

(**a**)　　　　　　　　　　　(**b**)

**Figure 4.** (**a**) Hardware photo (the title on it means on-line monitoring system of ship exhaust) and (**b**) monitoring site.

The software interface was developed according to the method in Section 2.3 and displayed the emission data monitored by the equipment (Figure 5). This system automatically analyzed the time series measurements of a plume and calculated the emission factor and the FSC.

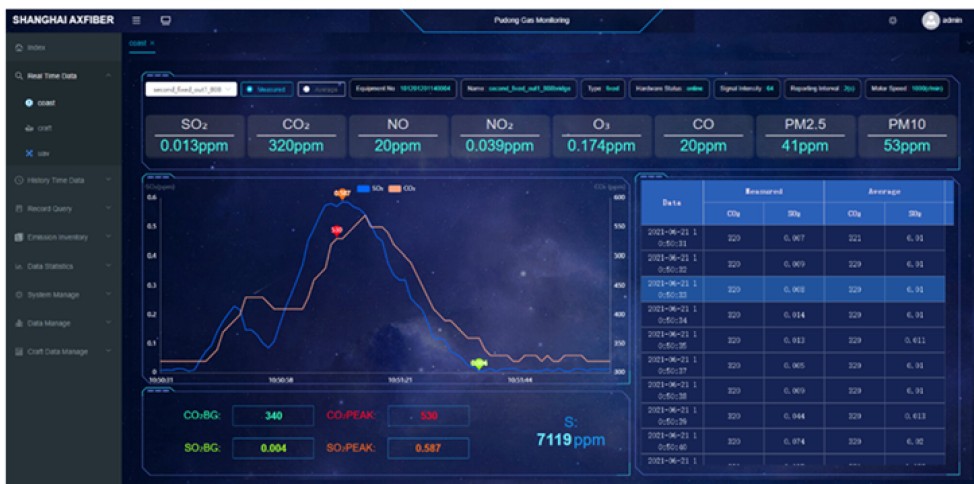

**Figure 5.** Software user interface.

The software also designated several sub-function modules, which retrieved detailed historical data for time or other fields. This statistical information could be used for summary reporting by providing emission inventory data, which could facilitate the understanding of the overall situation of the ECA. The system has powerful functionality, a simple and attractive interface, and straightforward operation, all of which could assist a maritime enforcement officer in understanding and mastering the environmental situation in the monitoring area, such as whether a ship's sulfur content exceeds regulations.

## 3. Experiments, Results, and Discussion

Ships can be divided by navigation area into ocean-going and inland river ships, which have different fuel requirements. According to the IMO convention, ships worldwide will not be allowed to use fuels with FSC $\geq$ 0.5% (m/m) after 2020. Inland river ships have relatively low emissions and are primarily regulated by local governments. In China, the 2018 implementation plan for ECAs of inland river ships requires an FSC of no more than 0.001% (m/m). To verify the practicability of the system, we chose two different locations to measure the emissions of seaports and river vessels, and analyzed the data measured to calculate the emission factors.

Since the beginning of 2020, we have used the system to measure the smoke plumes from two groups of ocean-going ships while they were berthed at Waigaoqiao Wharf in Shanghai, China. The smoke plumes measured included $CO_2$, $SO_2$, NO, $NO_2$, $PM_{2.5}$, and $PM_{10}$. As of March 2022, the equipment installed was stable and capable of 24/7 monitoring. Maritime supervisors can easily monitor a berthed ship as opposed to one that is sailing. Therefore, standard low sulfur fuel is commonly used. This resulted in relatively low ship emissions throughout the observation period. According to the ship berthing records, we chose two groups of data with the highest values as the standard data for analysis and calculation.

As shown in Figure 6, the measurements of two plumes were made on 21 June 2021. From the original data, we observed that when a ship was in the vicinity of the monitoring equipment, stable and accurate measurements were obtained. The method in Section 2.3 was then applied to calculate the emission factors of the ship exhaust gases and the PM.

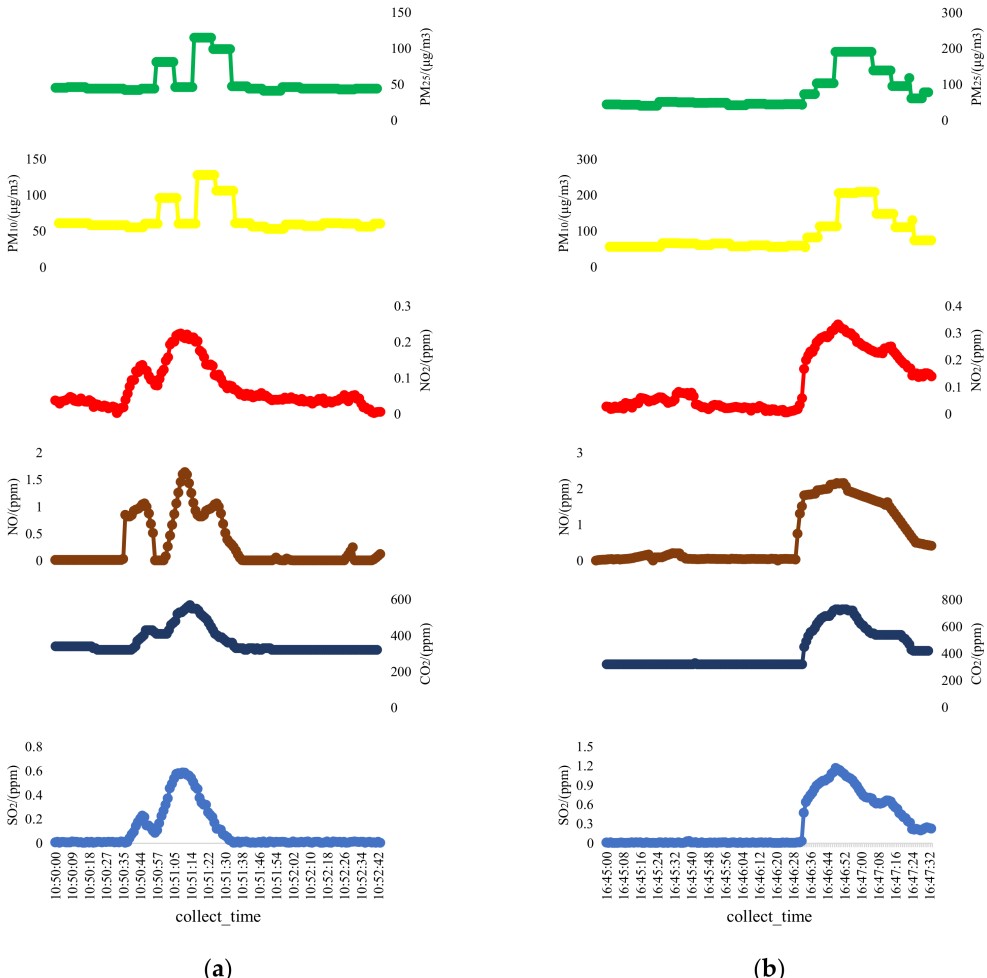

**Figure 6.** Measurements of typical plumes from two groups of ocean-going ships: (**a**) plume 1, (**b**) plume 2.

The EFs of plume 1 were: $SO_2$ = 10.53 g kg$^{-1}$, $NO_2$ = 2.88 g kg$^{-1}$, NO = 13.91 g kg$^{-1}$, $PM_{2.5}$ = 0.511 g kg$^{-1}$, and $PM_{10}$ = 0.518 g kg$^{-1}$, with an FSC of 0.53% (m/m). The EFs of plume 2 were: $SO_2$ = 12.87 g kg$^{-1}$, $NO_2$ = 2.59 g kg$^{-1}$, NO = 11.20 g kg$^{-1}$, $PM_{2.5}$ = 0.636 g kg$^{-1}$, and $PM_{10}$ = 0.644 g kg$^{-1}$, with an FSC of 0.64% (m/m).

Compared with ocean-going ship emissions, inland river ship emissions were lower. The monitoring location for the river-going ships was Gezhouba Lock no. 3, which is part of the Three Gorges Dam in the middle reaches of the Yangtze River, the largest river in China. Ships passing through the dam area need to be lifted or lowered through the lock before proceeding. Since March 2021, whenever a ship used the lock, our system measured the exhaust. We selected two typical plume data for comparison (Figure 7). We observed that although the emissions from inland river ships were small, the plume gas measurement data were still detected.

The EF results from plume 3 were: $NO_2$ = 12.08 g kg$^{-1}$, NO = 21.61 g kg$^{-1}$, $PM_{2.5}$ = 1.070 g kg$^{-1}$, and $PM_{10}$ = 1.968 g kg$^{-1}$. The results from plume 4 were: $NO_2$ = 9.12 g kg$^{-1}$, NO = 13.73 g kg$^{-1}$, $PM_{2.5}$ = 3.563 g kg$^{-1}$, and $PM_{10}$ = 4.399 g kg$^{-1}$.

According to the four sets of measurement data, we observed that all the measured values of gases and PM formed clear peak regions, except for $SO_2$ from inland river ships, which were used to calculate the emission factors and the FSC. According to the calculation results, the NO, $NO_2$ and PM emission factors of the inland river ships were greater than those of the ocean-going ships. This is mainly due to the fact that plumes 1 and 2 were measured when ocean-going ships were berthed, while plumes 3 and 4 were measured

when inland river ships passed through the gate at low speed. The NO emission factor was higher than that of $NO_2$ in both inland river ships and ocean-going ships, indicating that the monitoring location was close to the chimney and the measured NO had not been completely oxidized into $NO_2$ in the air. We observed that the $SO_2$ emission factors could be calculated for the ocean-going ships, but not for the inland river ships. This is mainly because the upper limit of FSC for inland river ships was 0.001% (m/m), which is far less than the 0.5% (m/m) for ocean-going ships. The FSCs of plume 1 and 2 were more than 0.5% (m/m), which indicated the possibility of using fuel with excessive sulfur content. The emission factor of particulate matter of the inland river ships was significantly higher than that of the ocean-going ships, which was mainly due to the slower speed of the ship when passing through the lock chamber and the idle state of the engine, which led to higher emissions of particulate matter for the inland river ships. The measurement data for the ocean-going ships were collected during monitoring while they were berthing.

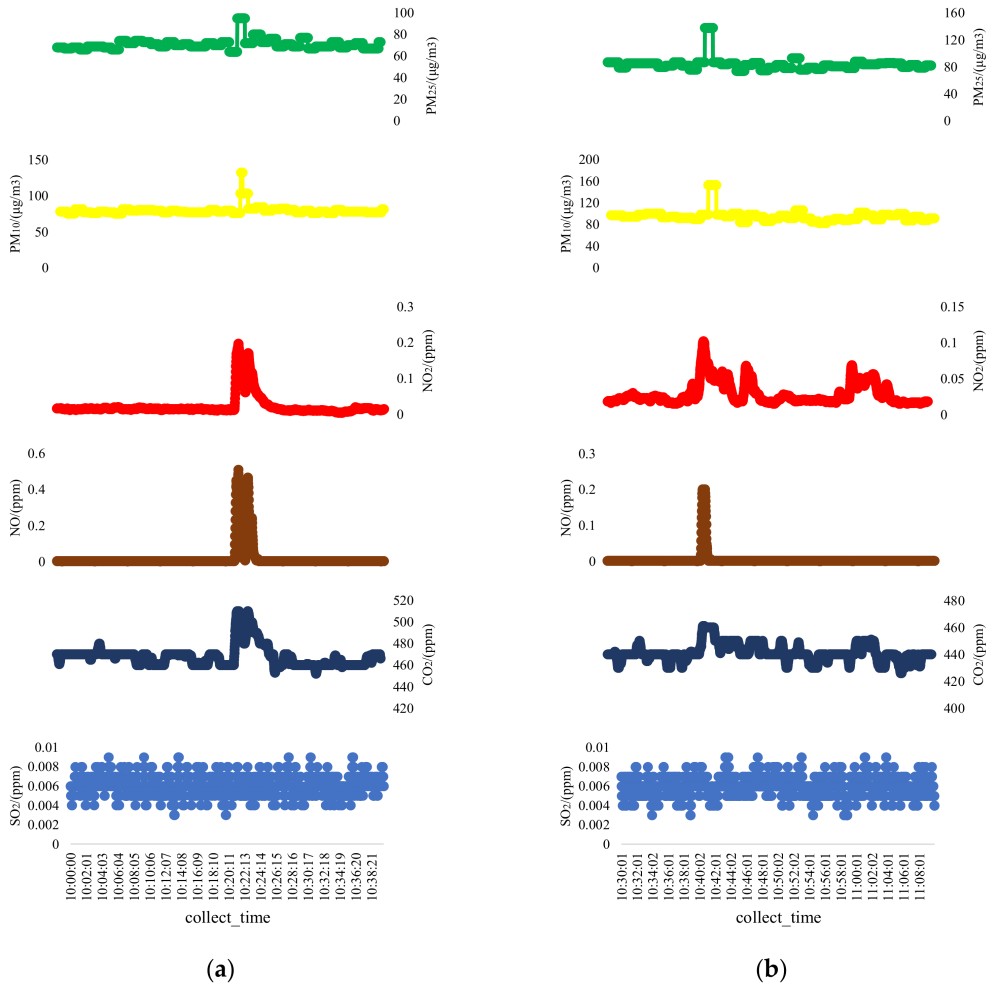

**Figure 7.** Measurement data of typical plumes from two groups of inland river ships: (**a**) plume 3, (**b**) plume 4.

## 4. Conclusions

In recent years, awareness of the harm caused by ship emissions to human health and the environment has grown. Although many researchers have undertaken research on ship emission monitoring, related equipment and methods have been primarily used for scientific exploration purposes. To effectively promote the implementation of ship emission control policies and aid the understanding of emission characteristics at different spatial and temporal scales, it is necessary to carry out comprehensive, real-time, and intelligent monitoring.

In this research, we designed and implemented a real-time measurement-modeling system for ship emissions that met these criteria. First, a portable device was developed, which could be mounted on a variety of maritime equipment and areas to carry out all-weather, all-day, and real-time ship emission monitoring. Then, based on a SSM framework and using Vue front-end technology, an information platform was developed. Finally, the emission factor of polluting gases and PM was calculated using a cloud server. Using this system, maritime enforcement officers can conveniently master the overall monitoring situation of ECAs and obtain real-time emissions of a target ship at the scene of maritime law enforcement.

To verify the validity of this system, we conducted experimental monitoring of ocean-going and inland river ships in Waigaoqiao Wharf and Three Gorges Dam, respectively. The results show that our system was capable of 24/7 emissions monitoring from these vessels, and the data collected were used to measure the emission factors of exhaust gases and PM. In the monitoring data, we selected four groups of typical data for analysis and calculation of emission factors. The nitrogen oxide emission factors for inland river ships and ocean-going ships had little difference. However, the FSC led to a difference in $SO_2$ emission factors, and the difference in flight status led to a difference in particulate emission factors.

In the future, our team will continue to study the refined calculation method for ship emission inventories and the application technology of remote environmental monitoring data in the maritime field, expand the application capacity of intelligent ship exhaust monitoring networks, provide additional technical support for the maritime department, and contribute to efforts for clean blue skies.

**Author Contributions:** F.Z. designed the study, analyzed the experimental data, and wrote the article. J.L. and X.Y. contributed to the experiments. H.Z. and Y.F. analyzed the experimental data. All authors have read and agreed to the published version of the manuscript.

**Funding:** This research was supported by the National Natural Science Foundation of China (grant No. 41701523) and Science and Technology Commission of Shanghai Municipality (grant No. 22692107400), as well as the Shanghai High-level Local University Innovation Team (Maritime Safety & Technical Support).

**Institutional Review Board Statement:** Not applicable.

**Informed Consent Statement:** Not applicable.

**Data Availability Statement:** Please address requests to Fan Zhou (fanzhou_cv@163.com).

**Conflicts of Interest:** The authors declare no conflict of interest.

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
