# Peer review of "A Real-Time Measurement-Modeling System for Ship Air Pollution Emission Factors"

_jmse, doi:10.3390/jmse10060760_

Round 1

Reviewer 1 Report

This paper has been well revised according to the reviewer's comments. The originality of the study was presented, and the methodology was well explained. Therefore, it is considered to have sufficient quality to be published in this journal.

Author Response

Thank you for your recognition of our work. We are greatly encouraged.

Reviewer 2 Report

Accepted but figures need to be improved

Author Response

Thank you for your recognition of our work. We are greatly encouraged.

I guess the figures need improved refer to the smoke plume measurements in FIG. 6 and FIG. 7. In the first submission, I used pictures, which were not very clear.  Therefore, I used vector images in the process of resubmission.  However, due to the size of the file, there seems to be a loss in the upload process.  I will submit the vector files separately later to ensure clarity if necessary. In addition, I modified the icon in FIG. 1. to make it more appropriate. 

Finally, thank you again for your positive suggestions.

Reviewer 3 Report

Authors provided an updated version of heir work that fulfills my previous comments. 

I am looking forward to read the next work that will includes parametric experiments and/or in-depth validation of the system. 

Author Response

Thank you for your positive comments on our work. We are very encouraged and will continue to do in-depth research in this direction.